# Influenza A Viruses in Whistling Ducks (Subfamily Dendrocygninae)

**DOI:** 10.3390/v13020192

**Published:** 2021-01-28

**Authors:** Deborah L. Carter, Paul Link, Gene Tan, David E. Stallknecht, Rebecca L. Poulson

**Affiliations:** 1Southeastern Cooperative Wildlife Disease Study, Department of Population Health, College of Veterinary Medicine, The University of Georgia, 589 D. W. Brooks Dr., Athens, GA 30602, USA; dlcarter@uga.edu (D.L.C.); dstall@uga.edu (D.E.S.); 2Louisiana Department of Wildlife and Fisheries, 2000 Quail Drive, Room 436, Baton Rouge, LA 70808, USA; plink@wlf.la.gov; 3J. Craig Venter Institute, 4120 Capricorn Lane, La Jolla, CA 92037, USA; gtan@jcvi.org; 4Division of Infectious Diseases, Department of Medicine, University of California, La Jolla, San Diego, CA 92037, USA

**Keywords:** avian influenza, Dendrocygninae, host, IAV, prevalence, surveillance, whistling ducks

## Abstract

As compared to other Anseriformes, data related to influenza A virus (IAV) detection and isolation, and IAV antibody detection in whistling ducks (*Dendrocygna* spp. and *Thalassornis leuconotus*; subfamily Dendrocygninae) are limited. To better evaluate the potential role of whistling ducks in the epidemiology of IAV, we (1) conducted surveillance for IAV from black-bellied whistling ducks (BBWD, *Dendrocygna*
*autumnalis*) sampled in coastal Louisiana, USA, during February 2018 and 2019, and (2) reviewed the published literature and Influenza Resource Database (IRD) that reported results of IAV surveillance of whistling ducks. In the prospective study, from 166 BBWD sampled, one H10N7 IAV was isolated (0.6% prevalence), and overall blocking enzyme-linked immunosorbent assay (bELISA) antibody seroprevalence was 10%. The literature review included publications and data in the IRD from 1984 to 2020 that reported results from nearly 5000 collected samples. For any given collection, the IAV isolation rate never exceeded 5.5%, and seroprevalence estimates ranged from 0 to 42%. Results from our prospective study in Louisiana are consistent with this historic literature; however, although all data consistently demonstrated a low prevalence of infection, the potential role of this species in the epidemiology of IAV should not be totally discounted. In sum, whistling ducks can be infected with IAV, they represent important species on many areas where waterfowl winter, and their distribution across the globe appears to be changing.

## 1. Introduction

Wild waterfowl in Order Anseriformes are important natural reservoirs for influenza A virus (IAV), but there have been limited reports of IAV in species of whistling ducks (WD; Family Anatidae, Subfamily Dendrocygninae). One of the eight currently recognized species of WD, the black-bellied whistling duck (BBWD; *Dendrocygna autumnalis*) is common in the Americas. BBWD breed throughout Louisiana, Texas, Mexico, and Central and South-central America, and form large flocks when not breeding, often in association with fulvous whistling ducks (FUWD; *Dendrocygna bicolor*) [1]. BBWD are largely residential apart from local movement, and tropical populations of BBWD are migratory with movement in the winter months [1]. The distribution of the species is expanding, and BBWD are now abundant along the southeastern coast of the United States [2]. Nocturnal feeders, BBWD forage on seeds, aquatic invertebrates, and vegetation in shallow freshwater [3].

In addition to the BBWD and FUWD, six other WD species are recognized worldwide in genus Dendrocygna, including *Dendrocygna arborea* (West Indian WD); *Dendrocygna arcuata* (wandering WD), *Dendrocygna eytoni* (plumed WD), *Dendrocygna guttata* (spotted WD), *Dendrocygna javanica* (lesser WD; LEWD), and *Dendrocygna viduata* (white-faced WD; WFWD); the white-backed duck (*Thalassornis leuconatus*) is also classified in the Dendrocygninae subfamily. Broadly, WD primarily exhibit only localized movements within their breeding ranges, with some exceptions. Small populations of LEWD in Asia have been noted to leave their northernmost breeding areas in the winter [4], and the globally widespread FUWD are migratory only in the most northern portions of their Mexican range [5].

Though infrequently sampled as compared to some other species in Order Anseriformes, exposure to and infection of IAV has been detected in resident and migrating Dendrocygninae, mostly outside of the United States. To better understand the role of WD in the ecology of IAV, we (1) conducted IAV surveillance of wintering BBWD along the Gulf coast of the United States over two years, and (2) reviewed published literature and the Influenza Resource Database (IRD) [6] for results from worldwide surveillance for IAV in this avian subfamily.

## 2. Materials and Methods

### 2.1. Virus Isolation and Antibody Detection 

Cloacal and oropharyngeal (COP) swabs and serum were opportunistically collected from live-captured BBWD in the Louisiana coastal parishes of Vermilion and Jefferson in February 2018 and 2019, respectively (Figure 1). In each year, approximately 40% of the sampled birds were after hatch year (AHY).

Collection sites included rice fields (2018) and a cargo ship terminal situated on the Mississippi River, where thousands of BBWD and other waterfowl were observed (personal observation, Paul Link, 2019). Virus isolation (VI) was performed on COP swabs as previously described [7]. Viral RNA was extracted (QIAamp Viral RNA Mini Kit, Qiagen Inc., Valencia, CA, USA) from all VI-positive samples following the manufacturer’s recommendation, and tested by IAV matrix real-time reverse transcriptase PCR (RRT-PCR) [8]. Whole-genome sequencing of IAV-positive isolates was carried out at J. Craig Venter Institute. Phylogenetic analyses were conducted using MEGA version 6 [9]; the top 2 NCBI basic local alignment search tool (BLAST) results, based on percent nucleotide (nt) similarity, were chosen on the basis of the criteria that they were from viruses identified from avian species at times predating the query sequence. Sterile blood samples were collected at a total volume less than or equal to 1% of the bird body mass via the brachial wing vein. Serum fractions were tested by nucleoprotein (NP) blocking enzyme-linked immunosorbent assay (bELISA; IDEXX, Westbrooke, ME, USA); samples with S/N ratios <0.7 were considered seropositive [10,11]. Animal work was approved by UGA Institutional Animal Care and Use Committee approval A2016 05-020-Y3-A6.

### 2.2. Literature Review

We searched over 3 decades (1984–2020) worth of publications on IAV surveillance in wild birds in subfamily Dendrocygninae: *D. arcuata*, *D. autumnalis*, *D. bicolor*, *D. eytoni*, *D. javanica*, *D. viduata*, and *T. leuconatus*. Prevalence data and metadata, including sampling country, species, sampling season, and year, and testing method(s) were extracted from each publication as available. Google Scholar (http://scholar.google.com) and PubMed (www.ncbi.nlm.nih.gov/pubmed, accessed on 1 September 2020) were searched from 1 September 2020 to 8 September 2020. Data were also obtained from the National Institute of Allergy and Infectious Diseases (NIAID) Influenza Research Database (IRD) [6] through the website of http://www.fludb.org, accessed on 15 October 2020; every attempt was made to assure that results reported from this outlet are not in publication elsewhere.

## 3. Results

### 3.1. Prospective—Virus Isolation and Antibody Detection

In total, 92 (2018) and 74 (2019) BBWD COP swab and serum samples were tested by both VI and NP bELISA. One H10N7 IAV was isolated in 2018 (A/black-bellied whistling duck/Louisiana/UGAI18-484/2018); no viruses were isolated in 2019. The gene sequences of the H10 hemagglutinin and N7 neuraminidase surface glycoproteins were most closely related to those of A/mallard/Ohio/16OS0620/2016 (H10N7) (Table 1). All internal genes were closely related to recently identified North American wild waterfowl sequences (Table 1). Three serum samples were NP bELISA-positive in 2018 (3.3% seroprevalence), and 14 in 2019 (18.9% seroprevalence), suggesting previous exposure to avian IAV. Seroprevalence in AHY birds across both years was 12.5% compared to 9.7% in hatch year (HY) birds.

### 3.2. Retrospective—Literature Review

As compared to some anatids, especially several *Anas* and *Anser* species, IAV surveillance has been fairly limited in Dendrocygninae, and often is the result of opportunistic sampling. Results of virus isolation (4942 samples) and/or PCR (4837 samples) testing were reported in 18 publications and include collection efforts of WD in Africa, Asia, Australia, and North and South America (Table 2). The overall PCR positivity rate was 1.7% (0–100% range), and the overall mean prevalence based on virus isolation was 0.1% (0–5.6%, range). In western Africa, 38 of 1269 (3.0%) whistling ducks tested IAV matrix-positive by PCR [12]. Cumming et al. [13] surveyed 165 avian species in southern Africa; IAV prevalence was 5.2% for birds in subfamily Dendrocygninae, twice as high as that reported from other ducks in the same study. In west Africa, highly pathogenic H5 genomes were identified in apparently healthy WFWD [14]. In the Americas, only three species of whistling ducks are known: BBWD, FUWD, and WFWD. Although infrequently sampled, IAV has been detected in each of these species. Influenza A viral RNA was detected in 14/489 (2.9%) BBWD and 5/71 (7.0%) WFWD in Colombia [15], and low pathogenicity H5N2 viruses were isolated from each of these hosts. In Texas, IAV was detected in 2/18 overwintering FUWD, resulting in the isolation of an H6N1 subtype IAV [16]. Five publications reported serological results from WD. Primarily on the basis of ELISA testing, we found that seroprevalence ranged from 0 to 42% (Table 3).

In addition to the published literature, the IRD included 4248 IAV results for WD sampled from 2007 to 2019 (Table 4). The majority of WD surveillance submissions to IRD (3655, 86%) were from LEWD sampled on the Asian continent. Of these, 223 (6.1%) were reported as IAV-positive.

## 4. Discussion

Results from efforts reported here indicate that BBWD are exposed to (seroprevalence range 3.3%–18.9%) and can be infected by (isolation of H10N7) avian IAV; the observed prevalence of IAV infection as determined by virus isolation (0.6%) was found to be low but consistent with prevalence estimates for other wintering duck species elsewhere in the Americas [7,33,34,35]. However, the NP antibody prevalence described in this study was substantially lower than that observed in mallard and teal species collected at sites in Louisiana at the same time of year (average 65%, 2018–2019; unpublished data). Low apparent seroprevalence in these resident WD may represent limited lifetime exposure to IAV due to this species’ limited geographic range within waterfowl wintering habitats in North America; this is supported by the low overall seroprevalence in adult (AHY) BBWD in this study (12.5%, average). Unlike mallard and teal species, WD are not present on traditional waterfowl breeding and staging areas during fall migration where IAV prevalence is consistently at its highest [34]. Disparate behavior and foraging strategies, largely localized movement, single species flocks, and differential susceptibility to IAV may also limit IAV exposure and play a role in the limited detection of antibodies we report in this resident species. Given that the ability of avian hosts to respond to exposure to IAV varies by species [36], differences in the ability of BBWD to mount a measurable immune response, as compared to other anatids, also cannot be discounted.

Travelling in large flocks and globally abundant on many waterfowl wintering areas, species of WD have been shown to be infected with similar IAV as migratory waterfowl. Further, the detection of highly pathogenic avian influenza (HPAI) virus in healthy WFWD in Africa [14] emphasizes the role that WD may have in the distribution and possibly the maintenance of HPAI and other pathogens of concern. Taken together, these results indicate that Dendrocygninae may play a role in the ecology and epidemiology of IAV, but given that they have been understudied in comparison to some anatid species, their potential importance in the natural history of IAV remains unknown. Results here, which are consistent with historic reports, demonstrate a low prevalence of infection in all WD species surveyed. Given that WD can be infected, the fact that they represent important species on many areas where waterfowl winter, and the fact that they have changing distribution in some parts of the world, the potential role of these species in the epidemiology of IAV should not be totally discounted.

## Figures and Tables

**Figure 1 viruses-13-00192-f001:**
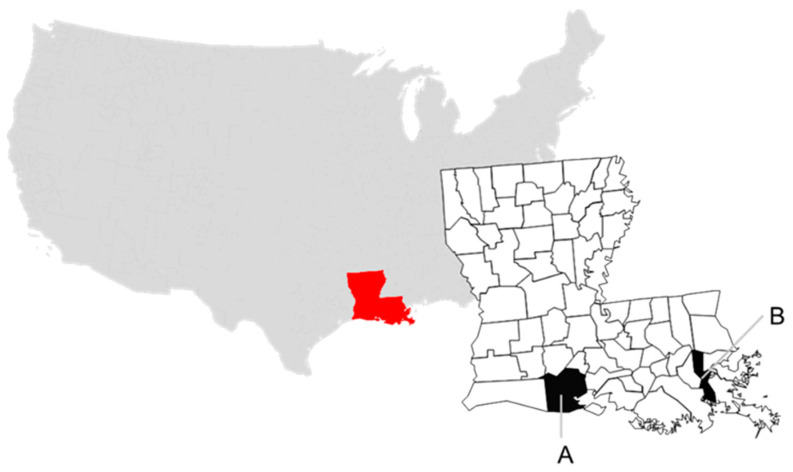
Collection sites in Louisiana, USA at parish level and by sampling year for this study: (**A**) Vermilion, 2018, and (**B**) Jefferson, 2019.

**Table 1 viruses-13-00192-t001:** Comparison of the eight influenza A virus (IAV) gene sequences for the virus isolated in this study (A/black-bellied whistling duck/Louisiana/UGAI18-484 (H10N7)) with avian IAV available on Genbank that have the highest nucleotide (nt) similarity. Results were obtained on 11 June 2020 using the NCBI basic local alignment search tool (BLAST) function; the top two BLAST results are listed.

A/BBWD/LA/UGAI18-484 (H10N7) Genbank Accession No. (Gene Segment)	BLAST Result		GenBank Accession No.	% nt Similarity
MN937687 (PB2)	First	A/MALL/IL/16OS4580/2016(H7N3)	MG279778	99.2
	Second	A/MALL/OH/17OS1892/2017(H10N7)	MK236922	99.1
MN937686 (PB1)	First	A/MALL/OH/16OS0620/2016(H10N7)	KY561093	99.4
	Second	A/COGO/WI/16OS4147/2016(H10N3)	MG280078	99.2
MN937685 (PA)	First	A/BWTE/WY/AH0099021/2016(H7N9)	MG266063	99.5
	Second	A/duck/AL/17-008643-2/2017(H7N9)	MF357835	99.5
MN937680 (HA)	First	A/MALL/OH/16OS0620/2016(H10N7)	KY561022	99.4
	Second	A/NSHO/NV/D1615770/2016(H10N9)	MK928244	99.1
MN937683 (NP)	First	A/GWTE/IL/17OS1400/2017(H7N3)	MG279995	99.5
	Second	A/MALL/OH/17OS1739/2017(H3N8	MK237096	99.4
MN937682 (NA)	First	A/MALL/OH/16OS0620/2016(H10N7)	KY561241	99.5
	Second	A/MALL/IL/16OS4899/2016(H10N7)	MG279737	99.3
MN937681 (M)	First	A/LESC/IL/17OS1577/2017(H7N3)	MG280109	99.4
	Second	A/NSHO/IL/11OS5680/2011(H11N9)	CY166988	98.9
MN937684 (NS)	First	A/NSHO/CA/HS206/2015(H8N4)	KY983106	99.3
	Second	A/BWTE/Guatemala/CIP049H102-18/2011(H1N3)	KX960456	99.3

Abbreviations were used in strain names for host species (BBWD = black-bellied whistling duck; BWTE = blue-winged teal; COGO = common goldeneye; GWTE = green-winged teal; LESC = lesser scaup; MALL = mallard; NSHO = Northern shoveler) and sample locations (AL = Alabama; CA = California; IL = Illinois; LA = Louisiana; MN = Minnesota; NV = Nevada; OH = Ohio; WI = Wisconsin; WY = Wyoming).

**Table 2 viruses-13-00192-t002:** Results (PCR and/or virus isolation) from literature search, including 18 publications reporting whistling duck samples collected for influenza A virus (IAV) screening, from 1984 to 2020. Yellow highlighted rows indicate the detection of highly pathogenic (HP) IAV.

Testing Method	Reference	Study Year	Location	Season	Species	Number Screened	Number PCR Positive (%)	VI Positive (%)	Subtype (#)
rrt-PCR screen only	Hoque et al. 2015 [17]	2007–2010	Australia	year-round	*Dendrocygna arcuata*	10	0 (0)	n/t ^6^	
					*D. eytoni*	1892	17 (0.9)	n/t	H6 (3) ^5^
rrt-PCR screen, VI ^2^ of PCR (+)	Gaidet et al. 2007 [12]	2006	Africa	overwintering	*D. bicolor*	102	1 (1)	0 (0)	
					*D. viduata*	1157	38 (3.3)	2 (0.2)	H1N1 (1), H8N4 (1) ^10^
	Cumming et al. 2011 [13]	2007–2009	Africa (southern)	N/R ^1^	*D. bicolor*	4	2 (50)	0 (0)	H6 (1) ^4^
					*D. viduata*	229	10 (4.4)	0 (0)	H5 (1), H7 (5) ^4^
					*Thalassornis leuconotus*	1	0 (0)	0 (0)	
	Pereda et al. 2008 [18]	2006–2007	Argentina	May	*D. autumnalis*	1	0 (0)		
				N/R ^1^	*D. bicolor*	139	0 (0)		
				May–July	*D. viduata*	53	0 (0)		
	Marinova-Petkova et al. 2014 [19]	n/r	Bangladesh		*D. javanica*	2196	0 (0)		
	Hurtado et al. 2016 [20]	n/r	Brazil	captive	*D. autumnalis*	86	0 (0)	0 (0)	
					*D. viduata*	6	1 (16.7)	0 (0)	
	Karlsson et al. 2013 [15]	2010–2012	Colombia	resident	*D. autumnalis*	489	14 (2.9)	1 (0.2)	LP ^11^ H5N2 (1) ^10^
					*D. viduata*	71	5 (7.0)	1 (1.4)	LP H5N2 (1) ^10^
	Gonzalez-Reiche et al. 2012 [21]	2007–2010	Guatemala	resident	*D. autumnalis*	1	0 (0)		
	Gonzalez-Reiche et al. 2016 [22]	2010–2013	Guatemala	winter migration	*D. autumnalis*	2	1 (50)	0 (0)	
					*D. bicolor*	2	2 (100)	0 (0)	
	Gaidet et al. 2008 [14]	2007	Nigeria	Feb	*D. viduata*	9	1 (11.1)	0 (0)	HP ^12^ H5N2 ^5^
	Caron et al. 2011 [23]	2007–2009	Zimbabwe	N/R ^1^	*D. bicolor*	1	0 (0)	0 (0)	
					*D. viduata*	152	9 (6)	0 (0)	
PCR screen, VI of PCR (+) (pooled samples)	Ofula et al. 2013 [24]	n/r	Kenya	Afrotropical; October, migration	*D. viduata*	34 [pools] ^9^	2 (5.9) ^9^	0 (0)	
PCR and VI of all	Ferro et al. 2010 [16]	2006–2009	Texas, United States	overwintering (November)	*D. bicolor*	18	2 (11.1)	1 (5.6)	H6N1 (1) ^10^
					*D. autumnalis*	15	0 (0)	0 (0)	
PCR screen and/or VI	Negovetich et al. 2011 [25]	n/r	Bangladesh	N/R		nr	nr	0 (0)	
Pooled VI or PCR ^3^	Curran et al. 2015 [26]	n/r	northern Australia	August–November	*D. eytoni*	1786 ^7^	23 (1.6) ^7^	1 (0.06) ^7^	H6N1 (1) ^10^
					*D. arcuata*	13 ^7^	nr	nr	
VI only	Wongphatcharachai et al. 2011 [27]	2009–2010	Thailand	January, winter	*D. javanica*	nr	nr	nr	H12N1 (3) ^10^
	Mackenzie et al. 1984 [28]	1977–1978	western Australia	year-round	*D. arcuata*	94	n/a	0 (0)	
					*D. eytoni*	60		0 (0)	
	Keawcharoen et al. 2011 [29]	n/r	Thailand	breeding visitor	*D. javanica*	54	Nt	2 (3.7)	HP H5N2 (2) ^10^
					PCR totals ^8^	4837	84	1.7%	
					VI Results ^9^	4942	7	0.1%	

^1^ Not recorded (n/r). ^2^ Virus isolation (VI). ^3^ Samples pooled for VI (1992-2004) or PCR (2005-2009). ^4^ Subtypes determined by PCR of original material. ^5^ Subtype determined by sequencing of original material. ^6^ Not tested (n/t). ^7^ Not included in PCR totals. ^8^ Only counted if not pooled and clearly tested by PCR. ^9^ Only counted if not pooled and clearly tested by VI. ^10^ Subtype of isolate. ^11^ Low pathogenicity (LP). ^12^ Highly pathogenic (HP).

**Table 3 viruses-13-00192-t003:** Results (serology) from literature search, including five publications reporting whistling duck samples collected for influenza A virus antibody screening, from 1984 to 2020.

Testing Method	Reference	Location	Season	Species	Number Screened	Initial Screen (% Positive)	Second Screen (% Positive)
cELISA ^a^ screen, HI ^b^ of ELISA(+)	Hoque et al. 2015 [17]	Australia	year-round	*Dendrocygna arcuata*	8	1 (12.5)	n/r
				*D. eytoni*	1209	90 (7.4)	n/r
AGID ^c^ and bELISA ^d^	Brown et al. 2010 [30]	Argentina	May	*D. autumnalis*	1	0 (0)	0 (0)
			unknown	*D. bicolor*	45	1(2.0)	12 (27.0)
			January–February		14	1 (7.1)	5 (35.7)
			May–July		31	0 (0)	7 (22.3)
			May–July	*D. viduata*	9	0 (0)	0 (0)
cELISA screen, HI	Curran et al. 2015 [26]	northern Australia	August–November	*D. arcuata*	11	1 (9.1)	n/r
				*D. eytoni*	1806	757 (41.9)	n/r
	Hassan et al. 2020 [31]	Bangladesh	migratory; winter	*D. javanica*	n/r ^e^	5	n/r
ELISA	Daodu et al. 2020 [32]	Nigeria	unknown	*D. viduata*	4	0(0)	n/a ^f^
				*Total screened*	3130	851 (27)	

^a^ Competitive enzyme-linked immunosorbent assay. ^b^ Hemagglutination inhibition (HI). ^c^ Agar gel immunodiffusion (AGID). ^d^ blocking (b)ELISA. ^e^ not reported (n/r). ^f^ not applicable (n/a).

**Table 4 viruses-13-00192-t004:** Results from Influenza Research Database search, not previously included in publications, reporting influenza A virus surveillance data spanning the years 2007–2019 for whistling duck species sampled on five continents.

	Continent					
Species	Africa	Asia	Australia	North America	South America	Total
*Dendrocygna autumnalis*	n/r ^1^	n/r	n/r	0/39	0/44	0/83
*D. bicolor*	0/8 ^2^	n/r	n/r	0/5	0/101	0/114
*D. eytoni*	n/r	n/r	0/3	n/r	n/r	0/3
*D. javanica*	n/r	223/3655 (6.1)	n/r	n/r	n/r	223/3655 (6.1)
*D. viduata*	0/14	n/r	n/r	n/r	0/377	0/391
*Dendrocygna* spp.	n/r	0/2	n/r	n/r	n/r	0/2
Total	0/22	223/3657 (6.1)	0/3	0/44	0/522	223/4248 (5.2)

^1^ not reported (n/r). ^2^ number positive/number reported (% positive).

## Data Availability

The data presented in this study are available in Appendix A.

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
