# Peer review of "Influenza A Viruses in Whistling Ducks (Subfamily Dendrocygninae)"

_viruses, 2021, doi:10.3390/v13020192_

Round 1

Reviewer 1 Report

Dear authors,

As the majority of influenza surveillance and research studies focus on Anas spp., in particular the mallard, this article will be a valuable contribution to the understanding of the epidemiology and ecology of influenza A viruses in non-mallard species. I mainly have minor comments, please see below.

Major comments

Introduction, line 30-55: Besides BBWD migratory behaviour, include foraging behaviour, preference for marine habitats or freshwater, population size, aggregation behaviour with other duck species (which species, also Anas spp.?),  etc. in the introduction

Discussion, line 174: please be more specific than "Targeted seasonal surveillance of WD and increasing global sample sizes". What is meant by "seasonal surveillance"? Which geographic areas and periods should be targeted based on global population data on WD? What would be minimal sample sizes given the prevalence data presented here?

Minor comments

Abstract

line 10-12: consider to remove "molecular" in "virologic/molecular" as it relates to the virus and is therefore also "virologic", or specify "virologic" (E.g. "As compared to other Anseriformes, data on influenza A virus (IAV) detection, isolation and antibody detection in whistling ducks (Dendrocygna spp. and Thalassornis leuconotus; subfamily Dendrocygninae) are limited.")

Introduction

Consider to include some sentences on IAV in wild birds/ Anseriformes (e.g. transmission routes, virus tropism, low pathogenicity etc)

line 49: "as compared to other species in Order Anseriformes" I would think that the majority of Anseriformes species has been sampled infrequently and that there are few exceptions (e.g. mallard, green-winged teal, blue-winged teal), therefore I would suggest to rewrite this sentence

Material and Methods

Consider to replace heading "2.1 Virus isolation and Serology" with e.g. "Virus isolation and antibody detection" (or "Virology and Serology" but that may be too general)

line 59: Was there a reason why birds were sampled in February? Are they more accessible during this time of year? If so, the authors could consider to include this in the M&M.

line 60: what is known about age and gender of the sampled birds?

line 75: insert "(Basic Local Alignment Search Tool)" after abbreviation BLAST

line 75: replace "identity" with "similarity"

line 85: replace "surveillance for IAV" with "IAV surveillance"

line 92: "results reported from this outlet do not appear to be in publication elsewhere." or "have not been published elsewhere"?

Results

line 97: is "samples" equal to number of different birds?

line 103: serum samples: is there anything known about age distribution of birds of sera pos vs. sero neg?

Caption Table 1: replace "avian origin IAV" with "avian IAV" as origin may suggest here that the genetically most closely related viruses were ancestral viruses to the H10N7.

line 120-121: Whistling ducks belong to the family of the Anatidae and are also anatids. In addition, the vast majority of species belonging to the Anatidae has been undersampled and sampled opportunistically. For these reasons, please rewrite the first sentence.

line 121-122: for clarity consider to replace "virologic" testing with "virus isolation"; if "molecular" testing is equal to "PCR" then the use of PCR may be more specific. "PCR" is also used in the following sentence.

line 127: Is Dendrocygna a family? I understood that Anatidae is the family, and Dendrocygninae is the subfamily.

line 128: replace "anatid ducks" with "other ducks" (as whistling ducks are also anatids or ducks, and all ducks are anatids)

Table 2, last column Subtype: remove "0" as other cells have been kept blank if no subtype available. For consistency, column VI positive (%) replace 0 with  0(0).

Table 3: align last column Second screen (% positive)

Discussion

line 154: "due to this species limited geographic range". What about the age of the sampled birds? If biased towards juvenile birds, then a low(er)seroprevalence may be expected. Please include age vs. (low) seroprevalence in the discussion.

line 155: replace "would not be" with "are not"?

line 157: What is known about "behaviour and foraging strategies" of WD? Please include here and/or in the introduction.

line 166: include highly pathogenic avian influenza before first use of abbreviation HPAI (or replace with "HP" as elsewhere in the article HP has been used)

line 166: insert "HPAI viruses and other" before "pathogens of concern" 

line 169: replace "other" with "some" as majority of anatids are undersampled

Author Response

Please see the attachment, and thanks kindly for your suggestions

Reviewer 2 Report

Influenza A Viruses in Whistling ducks (Subfamily Dendro- 2 cygninae) by Carter, Link, Stallknecht, and Poulson

The reviewed manuscript is clear and well-written and describes a rigorous and comprehensive approach to understanding the role of lesser studied species in avian influenza A virus ecology. The study fills an important knowledge gap and provides valuable information necessary for understanding and predicting risk.

I only have minor comments and suggestions.

Line 52: Consider mentioning upfront that Feb is wintering in LA; maybe include “wintering” on line 52?

Line 64: Nice figure.

Line 69: Are WD primarily found in mixed or single species flocks? Might be relevant to risk.

Line 76: Maybe say criteria rather than a two-part criterion (avian + date)?

Line 97: Recommend clarifying something like “swab and serum samples from BBWD…”.

Line 124. Recommend adding ranges to the “overall” results and clarifying whether the overall is a mean or median.

Line 135: The authors might consider mentioning the sequences for WD viruses posted to the IRD and include an H3N8 from Germany, and two H5N2s from Nigeria. I assume the 2 H5N2s are the same as those reported in the Lit Report summary in Table 2, but the H3N8 subtype doesn’t seem to be reported elsewhere.

Table 2: PCR and VI summary results - recommend aligning summary results below the Table with Table/Column Headers and putting percentages in parentheses as in the rest of the columns for clarity and to standardize with Table 3.

Table 4. Recommend indicating in the table or the table heading that the first number is the number of positives, the second number is the number samples and the parenthetical number is the percent.

Line 158: Perhaps clarify that the listed reasons might limit IAV exposure rather than immunity rather than cause a limited immune response which is what I think the sentence is suggesting? Are there any specific behavior or foraging behavior strategies that might reduce exposure in WDs compared to mallards or other Anatidae?

Line 159: I’m not sure what is meant by a “transient resident.” Does this refer to local movements?

Line 165: Consider changing HPAI to HPAIV since the birds were healthy with no signs of disease.

Line 170: Rather than “unknown” especially with the addition of the current study, I would suggest the role of WDs is “understudied.”

Line 172: The authors might consider explicitly stating whether WD commonly occur primarily in single-species or multi-species flocks. If they are often found in multi-species flocks, do they commonly co-occur with migratory species? Being a bit more explicit on this topic might provide more insight on potential risk.

Author Response

Please see the attachment, and thank you for your helpful suggestions.
